# Electrochemotherapy: An Alternative Strategy for Improving Therapy in Drug-Resistant SOLID Tumors

**DOI:** 10.3390/cancers14174341

**Published:** 2022-09-05

**Authors:** Maria Condello, Gloria D’Avack, Enrico Pierluigi Spugnini, Stefania Meschini

**Affiliations:** 1National Center for Drug Research and Evaluation, National Institute of Health, 00161 Rome, Italy; 2Biopulse S.r.l, 00144 Rome, Italy

**Keywords:** multidrug resistance, electroporation, head/neck squamous carcinoma, breast cancer, gynecological cancer, colorectal cancer

## Abstract

**Simple Summary:**

Chemotherapy is becoming an increasingly difficult antitumor therapy to practice due to the multiple mechanisms of drug resistance. To overcome the problem, it is possible to use alternative techniques, such as electrochemotherapy, which involves the simultaneous administration of the electrical pulse (electroporation) and the treatment with the drug in order to improve the effectiveness of the drug against the tumor. Electroporation has improved the efficacy of some chemotherapeutic agents, such bleomycin, cisplatin, mitomycin C, and 5-fluorouracil. The results of in vitro, veterinary, and clinical oncology studies are promising on various cancers, such as metastatic melanoma. The purpose of this review is to give an update on the state of the art of electrochemotherapy against the main solid tumors in the preclinical, clinical, and veterinary field.

**Abstract:**

Electrochemotherapy (ECT) is one of the innovative strategies to overcome the multi drug resistance (MDR) that often occurs in cancer. Resistance to anticancer drugs results from a variety of factors, such as genetic or epigenetic changes, an up-regulated outflow of drugs, and various cellular and molecular mechanisms. This technology combines the administration of chemotherapy with the application of electrical pulses, with waveforms capable of increasing drug uptake in a non-toxic and well tolerated mechanical system. ECT is used as a first-line adjuvant therapy in veterinary oncology, where it improves the efficacy of many chemotherapeutic agents by increasing their uptake into cancer cells. The chemotherapeutic agents that have been enhanced by this technique are bleomycin, cisplatin, mitomycin C, and 5-fluorouracil. After their use, a better localized control of the neoplasm has been observed. In humans, the use of ECT was initially limited to local palliative therapy for cutaneous metastases of melanoma, but phase I/II studies are currently ongoing for several histotypes of cancer, with promising results. In this review, we described the preclinical and clinical use of ECT on drug-resistant solid tumors, such as head and neck squamous cell carcinoma, breast cancer, gynecological cancer and, finally, colorectal cancer.

## 1. Introduction

One of the goals of cancer research is to identify new therapeutic systems that overcome the multiple mechanisms of drug resistance that can be observed both directly and after the early stages of patient treatment. For this reason, there has been a growing interest in identifying chemicals with chemosensitizing activity that can increase the efficacy of chemotherapeutic agents currently in clinical use. Agents recognized as chemosensitizing can be used both in adjuvant and synergistic therapy and in combination with chemotherapeutic agents to increase their intracellular accumulation and efficacy [1]. The ideal chemosensitizer should completely reverse the resistance phenomenon with no or few side effects on normal tissues. Unfortunately, this is not always the case and research must continue to improve and increase patient life. In this regard, physically facilitating drug delivery systems based on ultrasonic, electrical, and magnetic modulations are emerging [2]. Electrochemotherapy (ECT) is an innovative medical strategy to improve the uptake and efficacy of lipophobic anticancer agents, such as bleomycin or cisplatin. ECT is applied as a locoregional therapy combining the administration of anticancer agents with poor membrane permeability with electrical pulses (electroporation, EP) of appropriate characteristics (shape, voltage, frequency) [3]. In addition to allowing the use of a lower drug concentration, ECT allows localized therapy toward a specific molecular target, thus limiting side effects and improving tumor control [4].

This review will include, in addition to an overview of drug resistance and the technique of ECT, a brief review of the main human cancers in which this technique is currently used, such as squamous cell carcinoma of the head and neck, breast, gynecological, and colorectal cancer. The authors’ aim is to broaden the vision and focus on the possibilities that this technique could have for improving the solid tumor treatment.

## 2. Overview on Tumor Resistance Mechanisms

Chemotherapy is one of the strategies used in the fight against cancer; despite the progress made in recent years, there are many factors that interfere with the effectiveness of chemotherapy and determine its failure [5]. Drug resistance can be intrinsic if cancer cells are naturally resistant or acquired if it occurs following drug treatment. Cancer cells can show a different resistance to structurally and functionally different drugs; in this case, we refer to multi-drug-resistance (MDR). MDR is a complex and multifactorial phenomenon that involves different cellular structures and functions [6]. Drugs are extruded from the cells and their intracellular concentration falls below the cytotoxic threshold. Cancer is characterized by a dynamic cell pattern; with the evolution of the disease, the population becomes heterogeneous, including some genetically distinct subpopulations with different levels of sensitivity to treatments [7]. Tumor heterogeneity is a feature of aggressive tumors with different phenotypic and molecular properties both at the intertumoral and intratumoral levels [8]. These peculiar cellular characteristics very often lead to treatment failure. The main mechanisms involved in the acquisition and maintenance of resistance are summarized in Figure 1.

Many drugs, such as cisplatin, induce cancer cell death by binding to DNA; once DNA is damaged, cells can repair the lesions by multiple mechanisms and activate resistance mechanisms [9].

Metabolic resistance is the ability of the tumor cells to recognize the drug and chemically alter it through the expression of enzymes that metabolize it. Enzymatic activation of the drug leads it to exert its cytotoxic action directly in the tumor cell. Conversely, when enzymatic inactivation occurs, reduced pharmacological functionality is observed, thus protecting the cancer cell from damage [10].

The resistance of the drug target is characterized by the selective alteration of the interaction between the drug and its cellular target where it exerts the cytotoxic action, such as DNA, topoisomerase, metabolic enzymes, and components of the cytoskeleton. Drug-target interaction involves the selective recognition of the drug-specific binding site on the target molecule. If the binding site of the enzyme is mutated, poorly expressed, or absent prior to treatment, intrinsic resistance is triggered. Alternatively, cells may attempt to repair the damage or balance the loss by inducing increased expression of repair enzymes, and induced resistance is achieved [11].

Alteration of the cellular redox state is strongly involved in MDR. Many chemotherapy drugs induce cell death by apoptosis increasing the intracellular content of reactive oxygen species (ROS). Oxidative stress leads to the activation of selective macroautophagy, a process involved in the elimination of damaged organelles, such as defective mitochondria (mitophagy) [12,13]. Autophagy may act as an antioxidant defense mechanism by increasing cell survival and attenuating apoptotic cell death. Other pro-oxidant drugs may induce acquired chemoresistance because they are able to stimulate antioxidant defense by upregulating the transcription factors NFr2, NF-kB, FOXO, p53 [14].

The tumor microenvironment in which the tumor develops is a crucial determinant for tumor progression [15]. The crosstalk between tumor cells, stromal cells, and extracellular matrix favors the formation of a hypoxic, acidic, and immunosuppressive microenvironment [16,17,18]. The consequences of this crosstalk are tumor growth, metastasis, stem cell maintenance, relapse, and drug resistance. The expression of genes associated with stemness is crucial for tumor maintenance and may be mediators of resistance [19].

In recent years, a novel mechanism of drug resistance mediated by extracellular vesicles (EVs), such as exosomes or oncosomes, has been discovered. The drug resistance phenotype involving EVs is associated with cellular dedifferentiation and transformation into epithelial-mesenchymal transition cells and the adoption of a typical cancer stem cell phenotype [20]. EVs are cell-derived vesicles involved in signaling between cells and, by releasing their contents, can influence processes in target cells [21]. Exosomes have various molecules within them, such as proteins, lipids, metabolites, RNA molecules, and drugs [22,23]. Specific exosomes secreted by cells can be used as biomarkers to predict the presence of a tumor in patients [24].

Cancer cells activate several molecular mechanisms that allow them to resist death by apoptosis. Among the molecules that can activate cell survival signals are PI3K/AKT/PTEN, NF-kB, and Ras/Raf/MAPK [25].

The reduced intracellular drug concentration, typical of the MDR phenotype, may be due to either increased efflux or reduced accumulation of the drug in the target cell. These mechanisms are mediated by members of the ATP-binding cassette (ABC) superfamily. These are transport proteins, most of which act as efflux pumps—exporters. ABC proteins are classified into seven families (ABCA-ABCG), but among all members, only three (ABCB, ABCC, and ABCG) are involved in the active efflux of anticancer drugs out of the cell. The most studied transporters are P-glycoprotein (P-gp, ABCB1, or MDR1), multidrug resistance-associated proteins (MRP), and breast cancer resistance-associated proteins (BCRP, ABCG2) [26].

Although these mechanisms are different, they all contribute to the transformation of the normal cell into a resistant cancer cell. Numerous agents, such as curcumin and valspodar, have been used to inhibit P-gp activity. However, they often cause chronic toxicity and systemic side effects [27,28]. Today, strategies to overcome MDR are being evaluated, including bypassing P-gp-mediated drug efflux with nano-drug delivery systems (NDDS), such as nanotubes, micelles, liposomes, and metallic nanomaterials [29,30]. NDDS exploit clever mechanisms to achieve drug release under certain circumstances (such as under pH changes or hypoxia conditions) or drug accumulation in tumors through passive targeting (by enhanced permeability and retention, EPR) or active targeting (ligand–receptor binding), but they are still mostly in the preclinical stage. In this regard, ECT greatly increases the uptake of the drug, as bleomycin, thereby preventing tumor cells from removing the drug from the intracellular compartment and inhibiting DNA repair capacity. Many studies have demonstrated on in vitro models the enhanced efficacy of chemotherapeutics, such as doxorubicin, cisplatin, and vinorelbine, when administered with EP [4,31,32]. The authors suggested that EP modulates MDR by increasing cell membrane permeabilization and reducing intracellular ATP levels. Consequently, the increased drug efficacy with ECT allows for dose reduction in chemotherapy protocols. In the work of Condello et al., 2021, ECT was shown to inhibit mitomycin D drug-induced autophagy-mediated resistance [33]. Recently, ECT was successfully used to treat multi-drug resistant synchronous metastatic solid tumors in an elderly patient, thus preventing the use of radical surgery [34]. ECT is an established therapy and should be used in carefully selected patients to be effective in local lesion control and improve therapeutic planning.

## 3. The Electrochemotherapy History

The treatment of cancer patients, due to acute and delayed side effects, often leads to the interruption of radiotherapy and chemotherapy [35,36]. Preclinical studies have demonstrated in animal models and in various histological tumors an excellent anticancer efficacy of ECT, minimal toxicity, and safety of the procedure. In 1973, Crowley and colleagues began their work on lipid bilayers; in 1982, Neuman successfully performed the first genetic manipulation by electroporation; and, finally, in 1992, Okino introduced the term ‘electrical chemotherapy’ [37,38,39]. Subsequently, several studies were conducted to establish the appropriate electrical parameters to define the most effective ECT protocol: size, shape, electrode composition, electric field strength, shape and duration, frequency, and number of applied pulses important in the therapeutic response. The results showed that EP does not depend on length, but rather on the shape and width of the pulses and the size and porosity of the cell membrane [40]. Researchers initially used square electrical pulses with electroporation values below the permeability threshold to avoid cell damage. The first human clinical phase I and II studies performed in 1993 with square-wave EP demonstrated that ECT is effective [41]. This finding is strongly supported by subsequent studies showing that ECT can be used to treat primary or recurrent tumors of different histology. In 1999, Daskalov and colleagues demonstrated that the application of 8 pulses as a single burst of 50 + 50 µs biphasic pulses with a total duration of 7.1 msec, compared with a succession of 8 separate stimuli, was well tolerated by patients [42]. Biphasic pulses have a higher permeabilizing efficacy due to the inhomogeneous alignment of tumor cells with respect to field polarity. In 2014, Spugnini and colleagues obtained clinical proof of the efficacy and low toxicity concept of a new ECT protocol based on the use of 8 biphasic pulses with a voltage of 1300 V/cm and a duration of 50 + 50 µs with reduced interpulse interval (Figure 2B) [43]. This electroporator differs from commonly used instruments in that it allows for square waveforms delivered in a sequence of single pulses, thus prolonging the duration of the treatment (Figure 2A).

Studies have also been conducted on the discomfort to the animals during the treatment; contraction tests have shown less morbidity for animals treated with the new parameters. The application of this new ECT protocol offers several advantages: reduction in perceived pain, reduction in anesthetic risks, and improvement in pain control during treatment [44].

The efficacy of ECT in adjuvant therapy has been validated in combination with several anticancer agents against cutaneous and subcutaneous malignant lesions with increased cytotoxicity [45]. In 2011, Spugnini and colleagues investigated a new approach to improve the effect of therapy against a type of melanoma that does not respond to conventional therapies, using EP to promote the uptake and efficacy of antisense molecules. The results confirmed enhanced internalization of the antisense probe [46].

## 4. The Principles of Electroporation and the Effect on Tumor Cells

Electroporation is a biophysical technique that induces an increase in the cell membrane permeability through externally applied pulsed electric fields (Figure 3).

Over the years, many experiments have shown that electric pulses induce two phenomena at the cellular level: an initial phase of pore formation followed by a phase of pore enlargement during the period of pulse delivery [47]. These initial pores are called ‘transient electropores’, which, after the disappearance of the electric field, reduce in size and stabilize to form so-called ‘long-lasting electropores’ [48]. Many molecules (larger than a few kilo Daltons) can pass through transient electropores of the cell membrane. Therefore, these molecules must be present in the extracellular medium in sufficient quantities to be effectively transported into the cell during the short period of transient electroporation. The transport (speed) of macromolecules is inversely proportional to their molecular weight and the final concentration of these particles inside the cell is far from equilibrium (intracellular concentration above the theoretical value). It can, therefore, be deduced that mechanisms other than simple diffusion through electropores are involved in the translocation process [49]. The ‘long electropore’ is effective for the transport of small or medium-sized substances that move by simple diffusion regulated by their intra- and extracellular concentration. The cell membrane is the main barrier that prevents the diffusion of lipophobic compounds. When electrical pulses are applied, two different phenomena can be observed: reversible electroporation (RE) and irreversible electroporation (IRE), both used in clinical practice (Figure 4) [3]. In both cases, cells are exposed to an electric field that depolarizes the membrane and alters the function of its lipid and protein constituents. Altering the stability of the membrane allows molecules, ions, and water to pass freely from the extracellular environment to the cytoplasm and vice versa [50]. RE occurs if the membrane returns to its normal state after exposure to the electric field, only if the cell is able to pump calcium ions out of the membrane or sequester them within the endoplasmic or sarcoplasmic reticulum [3]. RE can induce multiple DNA damages leading to apoptotic death and activation of the immune system [51]. In the case of IRE-induced damage, there is an alteration of cellular homeostasis with apoptosis and necrosis, due to an increase in intracellular calcium concentration [52]. IRE is used in clinical routine as a non-thermal form of ablation of soft tissue malignancies [53]. The discriminating factors are pulse width, number, duration, and frequency. In RE, the pulse width varies from 300 to 1300 V/cm, and the number of pulses is 8. In IRE, the amplitude is up to 3000 V/cm, and the number of pulses is up to 40. Exposure to electrical pulses causes chemical changes in lipids and modulation of membrane protein function, inducing aggregation of transmembrane proteins forming pseudo-channels, which contribute to increased membrane permeability [54].

Electroporation transiently alters ionic fluxes, and the restoration of homeostasis (ATP-dependent) is the crossroads between RE and IRE [52]. Pore formation begins with exposure to the electric field, water bonds stabilize the nano-sized pores, and ions and other molecules pass through the membrane. The EP process consists of five steps: induction, expansion, stabilization, release, and memory effect [55]. They occur in microseconds, milliseconds, seconds, and hours, respectively. The pulse amplitude induces membrane permeabilization and affects pore size, with secondary alteration of ion homeostasis. This condition alters the Na^+^ and K^+^ gradients, thus leading to membrane depolarization. As a result, there is a massive influx of Ca^2+^, which acts as a molecular ‘switch’ between RE and IRE [56]. EP provides transient electrical pathways in the cell, sufficient to create significant intracellular fields that modify the cytoskeleton, intracellular membranes, and vesicles, and the opening of mitochondrial permeability transition pores (PTPs) is observed, with increased lipid peroxidation [57,58,59]. Electropermeabilization is followed by increased water/ion flux with osmotic swelling and release of metabolites, ATP from the cytoplasm [60]. After EP, if the cell has sufficient ATP, it transfers Ca^2+^ into the sarcoplasm and endoplasmic reticulum and repolarizes [61,62]. Otherwise, ROS are activated causing death by apoptosis or necrosis [63].

Many in vitro studies have confirmed that EP increases the cytotoxicity of chemotherapeutic agents by inducing cell death in drug-resistant cells. Proteomic analysis after combination treatment (EP + chemotherapy) showed downregulation of several signaling pathways, such as MAPK and VEGF, which are involved in cell proliferation, differentiation, and migration [64]. The efficacy of combination therapy has also been shown to be dependent on the immune status of the tumor. In vitro and in vivo studies have shown that ECT elicits a greater response in immunogenic tumors, such as breast and colorectal cancer, than in less immunogenic tumors [65,66,67].

## 5. Clinical Applications

Electrochemotherapy is applied as a one-off treatment, with possible repetition after eight weeks. Under general or local anesthesia, the chemotherapy drug is administered intravenously, followed by the application of electric pulses to the tumor area [68]. Prior to treatment, the location and spread of the tumor are determined by imaging. Standard operating procedures and equipment are still under development for the treatment of tumor nodules accessible by fibroscopy, endoscopy, laparoscopy, or open surgery [69]. An endoscopic system has recently been developed to deliver electrical pulses to gastrointestinal tumors for the treatment of inoperable colorectal tumors [70]. Different sets of plate and needle electrodes are available for different nodule size, shape, and depth [71]. Figure 5 provides some examples of electrodes developed for different ECT applications.

Non-invasive plate electrodes are applied for tumors on the surface of the skin, while needle electrodes are used invasively with sufficient depth to penetrate deeper tumors [72,73]. In humans, needles are arranged in parallel rows with a 4 mm space between them, for the treatment of small nodules, or in a circular (hexagonal) arrangement recommended for larger nodules (>1 cm in diameter). Needle electrodes can be placed through the tumor tissue to a depth of 3 cm. In veterinary medicine, a depth of up to 7 cm has been achieved with ultra-sonographic guidance [74]. The penetration of the electric field depends on the distance between the electrodes and the distribution of the electric field. For deep penetration of the electric field, a high voltage and adequate distance between the electrode plates must be applied [75]. The electric field is greatest around the electrode and between the electrodes, decreasing rapidly outside them [76]. Therefore, if the tumor is greater than the distance between the electrodes, to effectively treat the whole tumor, it is important to establish the precise position of the electrodes and optimize the sequential application of the pulses. ECT is mainly used to treat skin metastases to palliate/reduce the volume of inoperable tumors. In veterinary medicine, it is also used as an alternative to radiotherapy for soft tissue neoplasms that are not completely excised [77]. ECT can be used for lesions resistant to chemotherapy and radiotherapy. It is indicated in patients with severe comorbidity or patients who cannot participate in all other treatments.

The procedure is simple and fast (90 min) and is associated with a short hospital stay; if necessary, multiple cycles of ECT can be repeated.

## 6. Clinical Trials

The first clinical trials conducted in Europe and the United States since the early 1990s followed different and non-homogeneous protocols; then in 2006, the results of the multicenter European Standard Operating Procedure of Electrochemotherapy (ESOPE) project with Cliniporator (IGEA, Italy) provided the clinical procedures for a standardized, efficient, and safe protocol for the administration of electric pulses and drugs for the local treatment of any type of skin cancer nodule [78,79]. ECT is now mainly used in Europe as adjuvant therapy for the treatment of tumors refractory to conventional approaches, such as radiotherapy or chemotherapy, or those that cannot be surgically removed because of their distribution and location. In addition, it can be used as cytoreductive treatment in an organ-sparing procedure. The patient undergoes several sessions of electrochemotherapy to reduce the size of the tumor mass before surgical resection. The use of ECT in humans was initially limited to local palliative therapy for skin metastases of melanoma [80]. The interest of this technique in oncology has led to the development of systems for the treatment of refractory and inoperable deep neoplasms. New electrode systems have been designed to facilitate intraluminal, endoscopic, or image-guided approaches to access tissues anywhere in the body, such as tumors in the small intestine, brain tumors, gynecological tumors, and tumors in the urinary tract. In the standard procedure, electrode insertion is guided by intraoperative imaging support, such as X-ray, ultrasound, or computed tomography to insert the electrodes according to the specific geometry of the patient’s tumor. These tools facilitate treatment planning, and it is possible to estimate the electric field coverage and to have precise indications about the optimization of electrode shape, voltage, and distance of each pair. It is very important to consider the location of the tumor, especially when it is close to the heart (e.g., liver metastases), to use ECT safely without interfering with the electrical activity of the heart. The use of ECT is contraindicated for patients with cardiac pacemakers or those with metal parts if the treatment area is in close proximity to the implanted device [69].

A large number of preclinical and clinical phase I and I/II studies have demonstrated the efficiency and safety of ECT (Table 1).

## 7. Treatment of Solid Tumors with ECT

### 7.1. Head and Neck Squamous Cell Carcinoma (HNSCC)

Malignant tumors of the head and neck can occur in the salivary glands, sinuses, and oropharynx. Globally, in 2020, it was estimated that about 600,000 new cases were diagnosed, and more than 330,000 deaths occurred annually, with men having a higher risk than women (2:1 to 4:1) of developing them [81]. Despite continuous improvements in surgery, radiotherapy, and chemotherapy, survival rates are still less than 50%. Disease-related and treatment-induced functional impairment (speech and swallowing), cosmetic disfigurement, and recurrent metastases have an incidence of approximately 10% at initial clinical presentation, with an additional 20–30% of patients developing distant metastases during the course of the disease [82]. More than 90% of these malignancies are squamous cell carcinomas that develop from the epithelial cells lining the mucous membranes of various districts, such as the mouth, throat, nose, sinuses, larynx (vocal cords), pharynx, salivary glands, and thyroid. Several transcriptomic and epigenomic studies have characterized the molecular profile of the different subtypes of HNSCC. They focused on the molecular signatures of HPV- and HPV+ malignancies for the identification of specific biomarkers to guide treatment selection. Treatment for head and neck cancer may include surgery, radiotherapy, chemotherapy, targeted therapy (e.g., EGFR inhibitors), and immunotherapy (e.g., pembrolizumab (anti-PD-1) monotherapy) [74]. These therapies can be used alone or in combination, but in most cases, they have side effects. In recent years, in vitro and in vivo studies have demonstrated the efficacy of ECT. A recent study on oral squamous carcinoma cells showed that the combination of electroporation and mitomycin C enhanced apoptosis by inhibiting autophagy. These data were confirmed in two veterinary patients demonstrating the efficacy and tolerability of electroporation [33]. Pre-clinical studies performed on radio-resistant human pharyngeal cancer cells have confirmed the enhanced cytotoxic effect of the combination of bleomycin with electrical pulses [83]. Several clinical studies are ongoing using ECT either as cytoreductive treatment (preoperative phase) to reduce tumor mass prior to therapy, or as primary treatment for early-stage tumors [84].

Clinical results show a promising role of ECT in the treatment of patients with primary skin cancer in the head and neck area who are unsuitable for surgery or chemotherapy/radiotherapy [83].

### 7.2. Breast Cancer

Breast cancer is prevalent worldwide and in women of all ages, with increasing rates in old age. Major risk factors include genetic predisposition, family history, radiation exposure, obesity, excessive alcohol consumption, reproductive history (age of first pregnancy), tobacco use, and hormone therapies used [85]. Histopathological analysis has shown that breast cancer is a heterogeneous disease, most consisting of epithelial tumors arising from the cells lining the ducts or lobules; less common are non-epithelial tumors of the supporting stroma (e.g., angiosarcoma, primary stromal sarcomas, phyllodes tumor) [80]. Breast cancer can be classified into four main molecular classes: Luminal A (ER+/PR+/HER2−/lowKi-67); Luminal B (ER+/PR+/HER2−/+/high Ki-67); HER2-overexpression (ER−/PR−/HER2+), and triple-negative breast cancers (TNBCs) (ER−/PR−/HER2−) [86]. The expression rate in tumor tissue of the estrogen receptor (ER) and progesterone receptor (PR) can be very different, leading to clinically relevant differences in chemosensitivity and response to hormone therapy. Conventional treatment of ER+tumors may include endocrine therapy, based on the use of selective ER modulators, such as tamoxifen, ER down regulators, such as fulvestrant, and inhibitors of key molecules involved in cell cycle regulation [87]. Humanized monoclonal antibodies (mAbs), such as trastuzumab, pertuzumab, and 19H6-Hu, which recognize and bind to specific epitopes on the HER2 extracellular domain, are currently approved for the treatment of HER + tumors [88]. Juen et al. used a novel technology: HER2-targeting ADC, antibody-drug conjugates with monomethyl auristatin improved antibody-drug (trastuzumab) transport, and internalization of the cytotoxic agent. It also observed a positive effect on HER2 xenograft tumor models in mice [89]. Chemotherapy still remains the first line of defence for triple-negative breast/TNBC tumors even in combination with endocrine therapies. Immunotherapy, using selective mAbs, is also highly effective for the treatment of TNBC disease. Treatment, unfortunately, has limitations, due to the development of resistance against endocrine therapy, chemotherapy, and targeted therapy, which increases the likelihood of recurrence and metastasis formation. Regarding targeted therapies, patients cannot often benefit from this type of therapy, as it is very difficult to design drugs that are selective for certain targets.

In 2019, proteomic analyses were performed with the aim of investigating the anti-tumor mechanisms of ECT on MDA-MB-231 cells and human TNBC, in combination with cisplatin, paclitaxel, and doxorubicin [64]. The data showed that EP+cisplatin induced a metabolic switch in resistant cancer cells from the glycolysis to the oxidative pathway. Furthermore, EP+cisplatin led to a downregulation of pathways involved in cell proliferation, differentiation, and migration, and an increase in ROS production, inducing apoptotic cell death. It was also shown that in the BRCA1-mutated cell line (HCC1937), the combination of the PARP inhibitor (Olaparib) with EP increased the efficacy of drugs. This effect was due to the inhibition of DNA repair induced by bleomycin or cisplatin.

In in vivo experiments, delayed tumor growth and thus a higher survival rate of mice was also observed [90]. Currently, there are no clinical studies on patients with primary breast cancer.

### 7.3. Gynecological Cancer

Gynecological cancers are neoplasms that affect women in the uterus (endometrium and cervix) and ovaries. According to the incidence of human disease, cancer of the cervix is the fourth most common cancer in women. The main predisposing factors include estrogen dominance (high levels of estrogen without or with low levels of progesterone); common conditions are obesity, diabetes, and late menopause. Therefore, the risk of developing cervical cancer is strongly dependent on reproductive, sexual, and behavioural factors. Cervical cancer develops in the lower part of the uterus. Almost half of all cases occur in women between 35 and 55 years of age. Endometrial cancer is the most common in the post-menopausal period, from 50 to 70 years of age [91]. Hereditary mutations in the BRCA1 and BRCA2 genes are present, but other risk factors are infertility, hormonal treatment for infertility, ovarian polycystosis, and endometriosis [92]. Surgical removal of neoplastic tissue is considered the standard treatment for these diseases and is the most effective in early-stage tumors. Generally, surgery is combined with adjuvant radiotherapy and carboplatin/paclitaxel-based chemotherapy, as the risk of tumor recurrence is between 25–30% of cases. Today, molecularly targeted therapies (mAb bevacizumab and PARP inhibitors) are also used, both as first-line treatment and in cases of relapse [93]. These therapies are often debilitating and ineffective. ECT is used to achieve local tumor control and to improve disease symptoms. ECT has been used on superficial tumors in patients with vulvar squamous cell carcinoma [94]. This evidence is also supported by in vitro studies showing increased sensitivity of chemoresistant human cervical cancer cells to cisplatin after electroporation [32]. Other in vitro studies showed that ECT could be an effective treatment in combination with bleomycin. A potentiating effect was observed in human ovarian cancer cells (MDAH-2774) expressing ER, in combination with hormonal (17β-estradiol-based) therapies [95]. ECT could also be used in patients with deep gynecological tumors. This claim is supported by evidence from an ongoing phase II clinical trial, which opened patient recruitment in 2021 to assess the efficacy, feasibility, and safety of ECT in this tumor type (Clinicaltrials.gov).

### 7.4. Colorectal Cancer

According to the incidence of the disease in men, colorectal cancer is the third most common cancer diagnosed for both sexes and the second most common cancer worldwide in terms of mortality. The risk of developing this type of neoplasm increases with age (over 50 years). Other risk factors are inherited mutations, caused by a defect in the MLH1, MSH2 or MSH6 genes, which are involved in DNA repair, and adenomatous polyposis (APC) genes. Common familial syndromes related to colorectal cancer include Lynch syndrome (hereditary non-polyposis colorectal cancer, or HNPCC) and familial adenomatous polyposis (FAP) [96]. Pathological conditions, such as chronic inflammatory diseases, prostate cancer, cystic fibrosis, exposure to abdominal radiation, and lifestyle (diet, physical inactivity, smoking, alcohol abuse), increase the risk of developing this neoplasm [97]. The sections of the colon with the highest incidence of cancer are the colon, rectum, and appendix. In most cases, this neoplasm develops from adenomatous polyps, which are considered precancerous forms resulting from the uncontrolled proliferation of cells in the mucosa. The likelihood of a colon polyp evolving into an invasive form of cancer depends on the size of the polyp; the larger the size, the greater the likelihood that the tissue will become cancerous. The prognosis in patients with colorectal cancer depends strongly on the degree of local invasion into the tissue, infiltration of neighboring organs, and the presence of metastases in the lymph nodes. If the cancer has penetrated the wall of the large intestine and spread to adjacent lymph nodes, adjuvant chemotherapy after surgical excision may increase survival. Chemotherapy options include 5-FU and leucovorin, oxaliplatin, or capecitabine, but neoadjuvant approaches, based on chemoradiation therapy, are also used to reduce the tumor mass prior to surgery. In recent years, targeted therapies and immunotherapies have been developed specifically to control advanced stages of cancer. Targeted drugs can be used in combination with chemotherapeutics or alone and are directed against key factors that promote cancer cell survival, such as vascular endothelial growth factor (VEGF), epidermal growth factor receptor (EGFR), BRAF inhibitors, and kinase inhibitors [98]. Due to the high immunogenicity of colorectal cancer, humanized recombinant monoclonal antibodies are currently used to explore the role of immunotherapy in different stages of colorectal cancer [99]. So far, this immunotherapy treatment is limited to advanced stages refractory to conventional chemotherapeutic approaches. These therapies are very expensive, have major adverse events, and do not prevent the emergence of resistance [100]. Considering all these limitations, ECT can be a valid and effective approach to treat deep solid tumors. Electroporation using biphasic pulse trains improved the cytotoxicity of doxorubicin on multi-resistant colon cancer in vitro and on in vivo models [4]. In vitro experiments showed that EP sensitized colon cancer cells to the camptothecin analogue, SN38, by changing the expression of ABC transporters and disrupting the integrity of the cytoskeleton. The chemosensitizing effect was due to the induction of oxidative stress and apoptotic/ferropoptotic cell death [101]. Promising results from in vivo experiments highlighted the importance of the use of ECT in the treatment of colon cancer, where electropermeabilization significantly increased the intratumoral content of bleomycin and cisplatin [102]. These results were supported by experiments on murine colon carcinoma cells and experiments performed on immunocompetent and immunocompromised mice, showing that the application of electrical pulses with chemotherapy drugs at a very low concentration enhanced the cytotoxic effect of chemotherapy, leading to tumor cell death and complete tumor regression [65]. Thanks to these great results and the innovation of new electrodes, consisting of an endoscopic system to deliver electrical pulses directly to the tumor mass, this therapy could be extended to clinical practice [103]. To date, three phase I and phase II clinical trials have started to recruit patients to evaluate the efficacy and safety of neoadjuvant electrochemotherapy in early and advanced/inoperable colorectal cancer [70].

## 8. Conclusions

The advantages of using ECT in the treatment of solid tumors certainly lie in the fact that the procedure is inexpensive and easy to perform. In addition, systemic side effects are low, so it can also be used in elderly patients or those in poor physical condition. However, there may be some contraindications, as it is known that a tumor larger than 3 cm in diameter has a poorer response to ECT than nodules smaller than 1 cm, so it is very important to evaluate the performance parameters and experimental protocols for each individual and disease. In addition, the role that an accurate study of tumor resistance mechanisms can play in the choice of a treatment pathway should not be underestimated. It is important to assess the real feasibility of using chemo/radiotherapy in combination with a minimally invasive technique, such as ECT. Certainly, ECT can be a valuable aid to improve treatment and also achieve positive results in reducing side effects induced by conventional treatments.

## Figures and Tables

**Figure 1 cancers-14-04341-f001:**
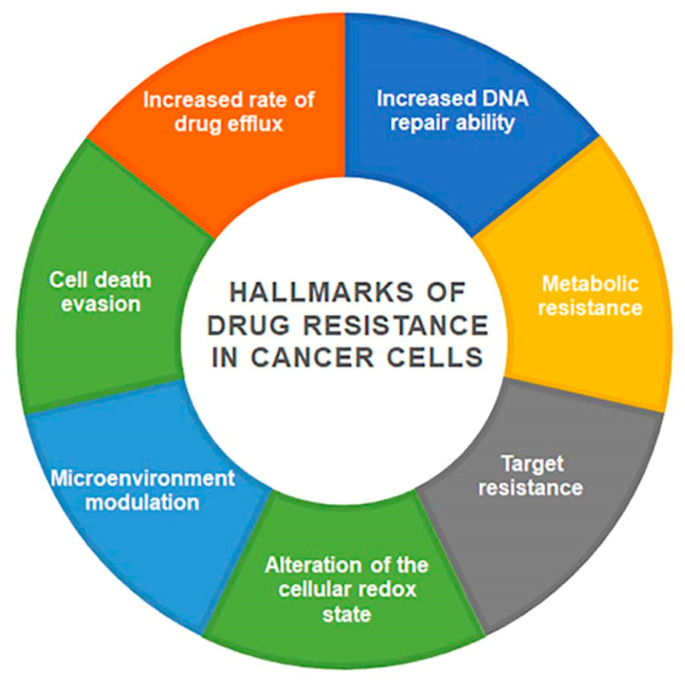
Schematic representation of the main mechanisms in MDR tumor.

**Figure 2 cancers-14-04341-f002:**
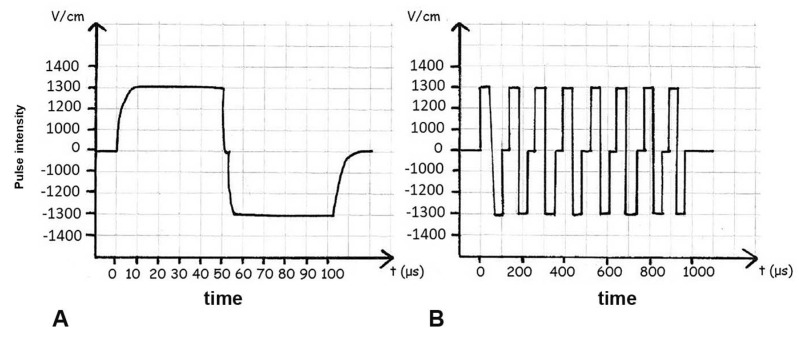
Different electroporation modes: (**A**) square waveforms delivered in a sequence of single pulses; (**B**) application of a series of 8 biphasic pulses, with voltage of 1300 V/cm and duration of 50 + 50 μs (time, t) each.

**Figure 3 cancers-14-04341-f003:**
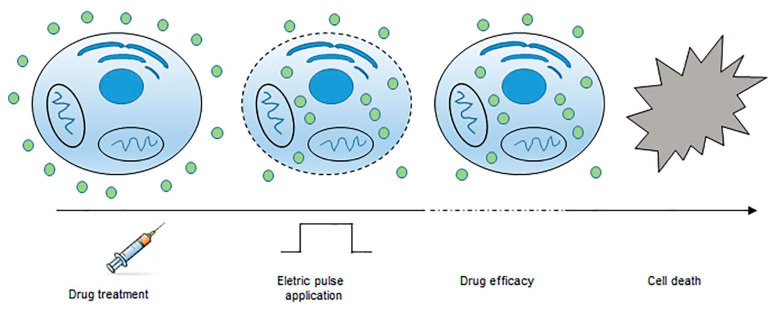
Principle of electrochemotherapy. After drug injection and application of intense and short electrical pulses, the cell membrane becomes more permeable. Most drug molecules enter the cells inducing cell death.

**Figure 4 cancers-14-04341-f004:**
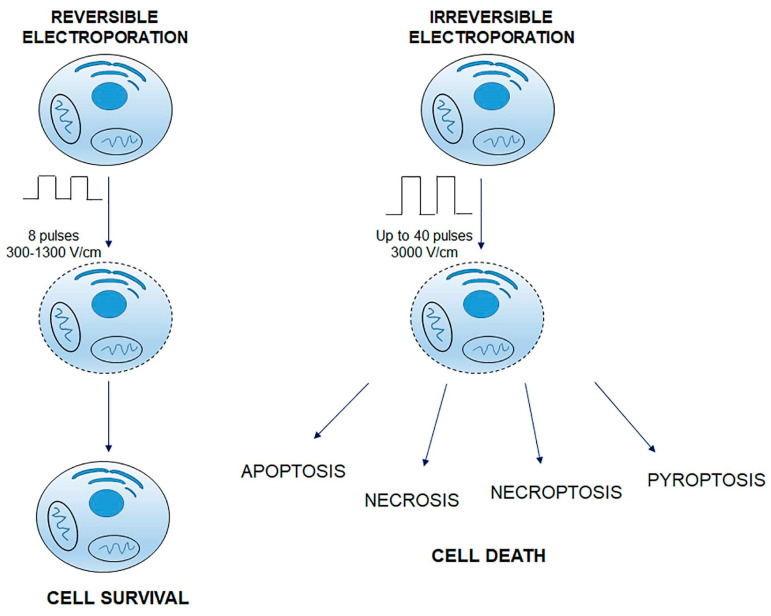
Reversible and irreversible electroporation. In both cases, the cells are exposed to an electric field. In RE, the pulse width varies from 300 to 1300 V/cm, and the number of pulses is 8; in IRE, the amplitude is up to 3000 V/cm, and the number of pulses is up to 40. In RE, after membrane permeabilization, cells recover and survive. In IRE, disruption of cellular homeostasis occurs; cell damage induces different types of cell death (apoptosis; necrosis; necroptosis; pyroptosis).

**Figure 5 cancers-14-04341-f005:**
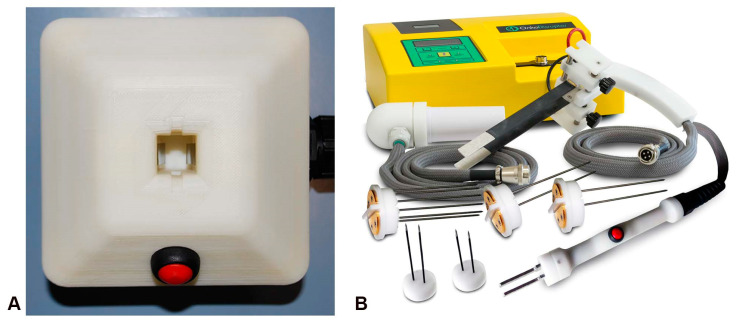
The main types of devices: (**A**) device for in vitro experiments; (**B**) pulse generator and needle electrodes for clinical use.

**Table 1 cancers-14-04341-t001:** Overview of clinical trials of electrochemotherapy.

Study Title	Phase	Interventions	Clinical Trials.gov Identifiers
Endoscopic-assisted Electrochemotherapy in addition to Neoadijuvant Treatment of Locally Advanced Rectal Cancer	II	Electrochemotherapy with bleomycin Device: EndoVE	NCT03040180
Electrochemotherapy for Non-curable Gastric Cancer	I	Electrochemotherapy with bleomycin	NCT0413907
Electrochemotherapy on Head and Neck Cancer	II	Electrochemotherapy with bleomycin Device: Cliniporator	NCT02549742
Electrochemotherapy of Posterior Resection Surface for Lowering Disease Recurrence Rate in Pancreatic Cancer (PanECT Study)	I/II	Electrochemotherapy with bleomycin Device: Cliniporator Vitae	NCT04281290
Electrochemotherapy of Gynecological Cancer (GynECT)	II	Electrochemotherapy with bleomycin or cisplatin	NCT04760327
Treatment of Primary Liver Tumors with Electrochemotherapy (ECT-HCC)	I/II	Electrochemotherapy with bleomycin Device: Cliniporator Vitae	NCT02291133
TMS Electrochemotherapy Glioblastoma Multiforme	II	Electrochemotherapy with temozolomide Device: TMS (Transcranial Magnetic Stimulation)	NCT02283944
Study of Folfirinox Electrochemotherapy in the Treatment of Pancreatic Adenocarcinoma	I	Electrochemotherapy with Folfirinox	NCT02592395
Electrochemotherapy for Chest Wall Recurrence of Breast Cancer: Present Challenges and Future Prospects	II	Electrochemotherapy with bleomycin Device: Cliniporator	NCT0744653

## Data Availability

Not applicable.

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
