# Peer review of "Electrochemotherapy: An Alternative Strategy for Improving Therapy in Drug-Resistant SOLID Tumors"

_cancers, 2022, doi:10.3390/cancers14174341_

Round 1

Reviewer 1 Report

The authors provide ample evidence of the utilization of electrochemotherapy (ECT) for the killing of tumors cells. The authors make it clear why ECT is an important adjuvant therapy for the treatment of cancers when they become resistant, primarily because it is cheaper alternative to treatment with expensive biologics.

There are only a few minor concerns for the authors to address, outlined below.

1.    Define the first instance of MDR (second line of abstract).

2.    While ECT is a cheaper alternative to treatment with expensive biologics, it is also invasive (in the clinic) with needles needing to be inserted to reach the site. Therefore, it seems that one of the main limitations of this type of therapy is that it should be to well localized tumors, and not disseminated (likely unknown ones)? Please comment on this.

3.    Furthermore, if electrodes need to be inserted to be effective and touch the tumor, why not just remove the tumor instead? Is this specifically for inoperable tumors? This should be highlighted as a significant advantage of ECT. Please clarify this in the “Clinical applications section”.

4.    How does ECT compare to other adjuvant/alternatives for MDR cancers?

5.    The “Overview on tumor resistance mechanism” seems out of place. It would help to add at least a few lines to clarify how ECT addresses tumor resistance mechanisms.

6.    Lines 108-112 about cancer stem cells. I would remove this section, as this is questionable, that the cancer stem cells or instead the rapid evolution of cancer cells evolving under pressure to acquire the resistance characteristics.

Reviewer 2 Report

Although there are many reviews available on this topic, it is a well-written manuscript. Minor grammatical corrections are needed. 

I would change the title since electrochemotherapy is not a new technique. The concept of electrochemotherapy is at least two decades old.  

Please revise section 3. You wrote the history of the technique instead of writing the principle of the technique.

Please add more figures to the manuscript. The manuscript looks bland as it is. Figures would make it interesting to the readers. 

Author Response

REVIEWER 2

Although there are many reviews available on this topic, it is a well-written manuscript.

Minor grammatical corrections are needed. We have corrected some grammatical errors.

I would change the title since electrochemotherapy is not a new technique. The concept of electrochemotherapy is at least two decades old.   We share the reviewer's thoughts. Electrochemotherapy is no longer a new technique, but it is certainly an alternative to classical chemotherapy. Therefore, we have changed new to alternative

Please revise section 3. You wrote the history of the technique instead of writing the principle of the technique. Section 3 describes the history of electrochemotherapy. Some principles of the technique have been described in the Section, where we have changed the title: "The principles of electroporation and the effect on cancer cells."

Please add more figures to the manuscript. The manuscript looks bland as it is. Figures would make it interesting to the readers. As suggested by the reviewer, we added some figures to make the manuscript more interesting for readers. The correct sequence is:

Figure 1. Schematic representation of the main mechanisms in MDR tumor.

Figure 2. Different electroporation modes. (A) square waveforms delivered in a sequence of  single pulses; (B)application of a series of 8 biphasic pulses, with voltage of 1300 V/cm and duration.

Figure 3. Principle of electrochemotherapy. After drug injection and application of intense and short electrical pulses, the cell membrane becomes more permeable. Most drug molecules enter the cells inducing cell death.

Figure 4 Reversible and irreversible electroporation. In both case the cells are exposed to an electric field. In RE the pulse width varies from 300 to 1300 V/cm and the number of pulses is 8; In IRE the amplitude is up to 3000 V/cm and the number of pulses is up to 40. In RE, after membrane permeabilization, cells recover and survive. In IRE, disruption of cellular homeostasis occurs; -; cell damage induces different types of cell death (apoptosis, necrosis; necroptosis; pyroptosis).

Figure 5. The main types of devices. (A) Device for in vitro experiments; (B) pulse generator and needle electrodes for clinical use.

Round 2

Reviewer 2 Report

The authors responded to my comments and improved the quality of their manuscript. It can be accepted in its current form.